# Refining Context-Entangled Content Segmentation via Curriculum Selection and Anti-Curriculum Promotion

Chunming He [1]   Rihan Zhang [1]   Fengyang Xiao [1]   Dingming Zhang [1]   Zhiwen Cao [2]   Sina Farsiu [1]

## Abstract

Biological learning proceeds from easy to difficult tasks, gradually reinforcing perception and robustness. Inspired by this principle, we address Context-Entangled Content Segmentation (CECS), a challenging setting where objects share intrinsic visual patterns with their surroundings, as in camouflaged object detection. Conventional segmentation networks predominantly rely on architectural enhancements but often ignore the learning dynamics that govern robustness under entangled data distributions. We introduce CurriSeg, a dual-phase learning framework that unifies curriculum and anti-curriculum principles to improve representation reliability. In the Curriculum Selection phase, CurriSeg dynamically selects training data based on the temporal statistics of sample losses, distinguishing hard-but-informative samples from noisy or ambiguous ones, thus enabling stable capability enhancement. In the Anti-Curriculum Promotion phase, we design Spectral-Blindness Fine-Tuning, which suppresses high-frequency components to enforce dependence on low-frequency structural and contextual cues and thus strengthens generalization. Extensive experiments demonstrate that CurriSeg achieves consistent improvements across diverse CECS benchmarks without adding parameters or increasing total training time, offering a principled view of how progression and challenge interplay to foster robust and context-aware segmentation. The code is available at https://github.com/ChunmingHe/CurriSeg.

---

[1]Duke University  [2]Adobe.   Correspondence to: Fengyang Xiao <fengyang.xiao@duke.edu>, Sina Farsiu <sina.farsiu@duke.edu>.

*Proceedings of the 43rd International Conference on Machine Learning*, Seoul, South Korea. PMLR 306, 2026. Copyright 2026 by the author(s).

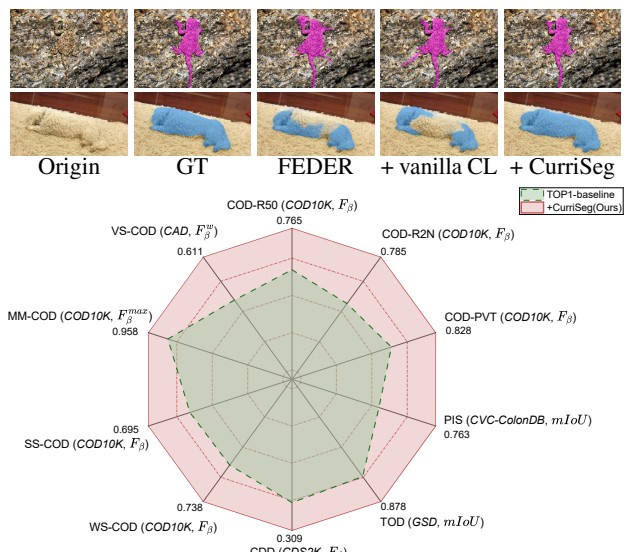

*Figure 1.* Performance on CECS data. **Top**: CL is short for curriculum learning (Bengio et al., 2009). Concealed object masks are highlighted in purple and blue, overlaid on the original data for visual clarity. **Bottom**: "Task (Dataset, Metric)". The radar chart shows that training the baseline under our CurriSeg brings performance gains than the standard training manner, where TOP-1 is the composite baselines with the top metric scores per task.

## 1. Introduction

Context-Entangled Content Segmentation (CECS) aims to segment those extremely concealed objects that share intrinsic visual patterns with their surroundings (Fan et al., 2020; Zhao et al., 2024; He et al., 2024b; Lu et al., 2024; Xiao et al., 2024; He et al., 2026), with applications in medical analysis (Tajbakhsh et al., 2015; Hu et al., 2026; He et al., 2025b), autonomous perception (Mei et al., 2020; Hu et al., 2025; Gao et al., 2025; Zheng et al., 2025), etc.

Unlike generic segmentation, CECS suffers from extreme feature ambiguity. Existing methods mainly focus on architectural engineering, such as feature aggregation (Pang et al., 2022; Fan et al., 2020) complex decoders (He et al., 2025e;c), or prior injection (Xiao et al., 2026b;a), yet overlook a fundamental bottleneck: the learning dynamics when facing entangled data distributions.

A natural strategy to handle such complexity is Curriculum Learning (CL) (Bengio et al., 2009), learning from easy to

hard, mimicking biological cognition. However, we argue that standard CL is detrimental in CECS scenarios. In camouflaged scenes, "easy" samples often exhibit high spurious correlations (*e.g.*, distinct background textures that unintentionally reveal the object). Training on these first drives the model into lazy regimes that rely on high-frequency texture bias rather than semantic structure, leading to poor generalization on hard samples, as validated in Fig. 1.

To address this, we introduce CurriSeg, a novel framework that orchestrates training through two phases: Robust Curriculum Selection and Anti-Curriculum Promotion. Inspired by biological mastery, where predators first acquire fundamental skills then adapt to challenging environments, CurriSeg stabilizes optimization before deliberately "blinding" the model to encourage deeper feature extraction.

In the Curriculum Selection phase, we address the noise–ambiguity dilemma inherent in CECS. Instead of relying on static difficulty measures, we monitor temporal behavior of samples. By tracking mean and variance of sample losses, CurriSeg distinguishes hard-but-informative samples from noisy or ambiguous ones. We also incorporate pixel-level uncertainty estimation to prevent uncertain regions from dominating early gradients. Combined with a warm-up curriculum strategy, this yields a joint image- and pixel-level curriculum that is stable and noise-resistant.

After the model reaches a stable regime, we introduce Anti-Curriculum Promotion to enhance robustness on hard samples. We propose Spectral-Blindness Fine-Tuning (SBFT), which deliberately attenuates high-frequency input components. By temporarily suppressing texture cues, SBFT forces the model to rely on low-frequency structural patterns and contextual semantics, promoting extraction of intrinsic, task-relevant signals. This explicitly counteracts the tendency to exploit superficial texture shortcuts and encourages discovery of complementary, subtler cues, improving robustness where texture-based heuristics fail.

Together, these two phases establish a principled paradigm that stabilizes optimization then intentionally increases difficulty to overcome representational bottlenecks. CurriSeg combines curriculum and anti-curriculum learning within a unified framework, offering a new perspective on training for context-entangled visual tasks.

Our contributions are summarized as follows:

**(1)** We propose CurriSeg, the first CL-based CECS paradigm. By following a "stabilize-then-perturb" trajectory, it effectively breaks the learning bottleneck caused by feature ambiguity in concealed scenes.

**(2)** We design robust curriculum selection that uses temporal loss statistics and uncertainty-aware pixel masking to construct a stable, noise-resistant learning trajectory.

**(3)** We introduce an anti-curriculum promotion phase, SBFT, which attenuates high-frequency cues and encourages the model to exploit complementary, subtle signals, thereby improving robustness on challenging camouflaged cases.

**(4)** We validate CurriSeg on multiple CECS benchmarks and backbones, showing consistent performance gains. CurriSeg adds no extra parameters and maintains training cost comparable to standard training schedules.

## 2. Related Works

**Context-entangled content segmentation.** Existing CECS methods mainly focus on structural designs: encoder-decoder backbones with multi-scale aggregation (Fan et al., 2020; 2021a; Li & Fu, 2025), attention-based refinement (Pang et al., 2022; 2024), edge- or uncertainty-aware branches (He et al., 2023a; 2025d), transformer-style context modeling (Sun et al., 2024; Zhang et al., 2025), and frequency-aware modules (He et al., 2025f; Shen et al., 2025). They enhance feature representation at the network level, yet training typically follows a standard supervised routine with shuffled mini-batches. The roles of sample difficulty, label ambiguity, and texture shortcuts in shaping CECS learning dynamics remain underexplored. Our work instead organizes the training procedure via curriculum and anti-curriculum mechanisms, orthogonal to existing architectural advances.

**Curriculum learning**. CL organizes training samples from easy to hard to improve generalization (Bengio et al., 2009; Soviany et al., 2022). Variants include self-paced learning (Kumar et al., 2010; Tullis & Benjamin, 2011), teacher–student curricula (Matiisen et al., 2019; Saglietti et al., 2022), and task-specific schemes via loss-based ranking (Pentina et al., 2015) or hardness-aware sampling (Soviany, 2020). However, CL assumes easy samples provide reliable supervision and difficulty correlates with learning utility. In CECS, these assumptions break down: "easy" samples may contain spurious textures promoting shortcuts, while hard samples can be both informative and ambiguous. Moreover, most CL methods overlook the pixel-level nature of segmentation with spatially localized uncertainty. CurriSeg departs from CL by (i) using temporal loss statistics to distinguish hard-but-informative samples from noisy cases, and (ii) adding an anti-curriculum phase that increases difficulty via spectral manipulation, mitigating texture bias.

## 3. Methodology

CurriSeg is a dual-phase learning framework designed to address the feature ambiguity inherent in CECS. As illustrated in Fig. 2 and Algorithm 1, the framework follows a "stabilize-then-perturb" trajectory comprising two sequential phases: (1) Robust Curriculum Selection (RCS), which constructs a stable, noise-resistant learning schedule through

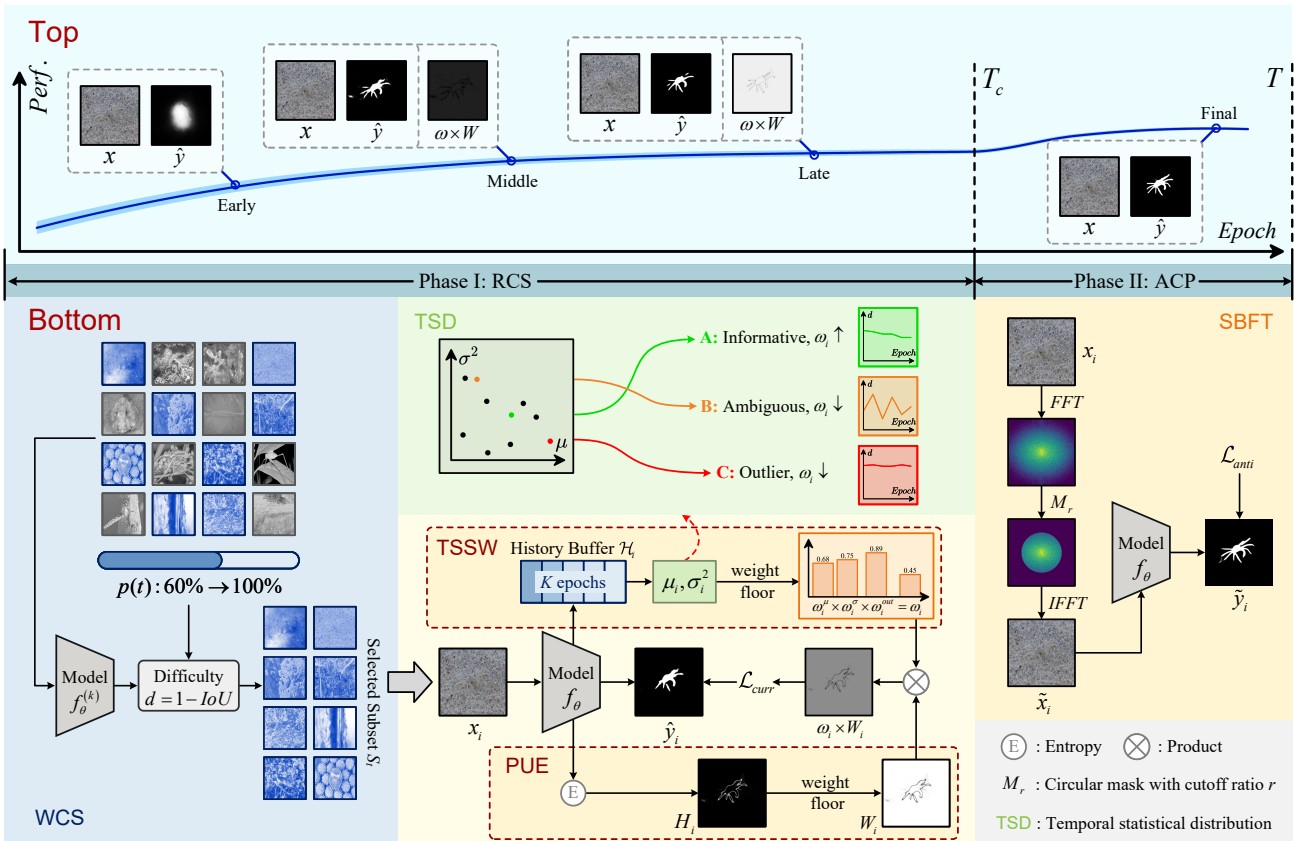

*Figure 2.* Framework of CurriSeg. In (1), performance improves steadily; brighter regions in $\omega \times W$ indicate more trustworthy pixels. In (2), Phase I applies Robust Curriculum Selection with sample/pixel-level weighting, while Phase II includes Anti-Curriculum Promotion via spectral filtering. TSD describes how to distinguish hard-but-informative samples from ambiguous or outlier ones.

temporal statistics monitoring and uncertainty-aware weighting, and (2) Anti-Curriculum Promotion (ACP), which deliberately attenuates high-frequency cues to encourage the model to exploit complementary, subtle discriminative cues.

## 3.1. Robust Curriculum Selection

This phase targets the noise–ambiguity dilemma inherent in CECS by monitoring temporal behavior of samples and incorporating pixel-level uncertainty, yielding a joint image- and pixel-level curriculum for stable optimization.

**Warm-up curriculum strategy**. Following (Bengio et al., 2009), we design a warm-up curriculum strategy (WCS) that progressively expands the training set using a model-centric difficulty measure based on segmentation performance. Concretely, we maintain a historical checkpoint $f_{\theta^{(k)}}$ saved every $K$ epochs (we set $K = 10$, so $k \in \{0, 10, 20, \cdots\}$). During the subsequent $K$ epochs, this checkpoint serves as an evaluator of sample difficulty. For each training sample $i$, we compute its difficulty score $d_i$, formulated as:

$$d_i = 1 - IoU(f_{\theta^{(k)}}(x_i), y_i), \tag{1}$$

where $f_{\theta^{(k)}}(x_i)$ is the predicted mask of the concealed input $x_i$ from checkpoint model, $y_i$ is the ground-truth, and

$IoU(\cdot)$ is the Intersection-over-Union. Samples with higher $d_i$ are considered more difficult for the current model.

We then implement a warm-up curriculum schedule by gradually expanding the training set. Let $p(t)$ denote the selection percentile at epoch $t$, and let $\mathrm{Per}(\{d_n\}_{n=1}^N, p(t))$ be the corresponding percentile threshold over all $N$ training samples. The active training subset at epoch $t$ is

$$\mathcal{S}_t = \{i \mid d_i \leq \mathrm{Per}(\{d_n\}_{n=1}^N, p(t))\},$$
$$p(t) = p_{min} + (1 - p_{min}) \cdot \frac{t-1}{T_c - 1}, \tag{2}$$

where $1 \leq t \leq T_c$ and $p_{min} = 0.6$. In practice, training starts from a subset of relatively easy samples (small $p(t)$) and progressively incorporates more difficult ones as $p(t)$ increases. This encourages the model to first consolidate its understanding of easier cases before being exposed to the full complexity and ambiguity dataset.

**Temporal statistics-based sample weighting**. Within the selected curriculum subset, not all samples contribute equally to effective learning. Due to object concealment, some samples are intrinsically ambiguous or affected by annotation noise; treating them on par with well-defined samples may destabilize optimization. To mitigate this,

**Algorithm 1** CurriSeg: Dual-Phase Curriculum Learning

---

**Require:** Dataset $\mathcal{D} = \{(x_i, y_i)\}_{i=1}^N$, model $f_\theta$, epochs $T$, curriculum phase $T_c$, number of training samples $N$
**Ensure:** Trained model $f_\theta$
1: Init $\theta$, history buffer $\mathcal{H}$, checkpoint $\theta^{(0)} \leftarrow \theta$
2: **for** $t = 1$ to $T$ **do**
3:    **if** $t \leq T_c$ **then**
4:    *// Phase 1: Robust Curriculum Selection*
5:       Compute difficulty $d_i = 1 - \text{IoU}(f_{\theta^{(k)}}(x_i), y_i)$
6:       Select $\mathcal{S}_t \leftarrow \{i \mid d_i \leq \text{Per}(\{d_n\}_{n=1}^N, p(t))\}$
7:       Compute sample weights $\{\omega_i\}$ from $(\mu_i, \sigma_i^2)$
8:       **for** each batch $\mathcal{B} \subset \mathcal{S}_t$ **do**
9:          Forward: $\hat{y}_i \leftarrow f_\theta(x_i)$
10:        Compute pixel weights $W_i$ via entropy
11:        $\mathcal{L}_{\text{curr}} \leftarrow \frac{1}{|\mathcal{B}|} \sum_i \omega_i \cdot (L_{BCE}^w(W_i) + L_{IoU}^w(W_i))$
12:        Update $\theta, \mathcal{H}$
13:       **end for**
14:       Save checkpoint if $t \mod K = 0$
15:    **else**
16:    *// Phase 2: Anti-Curriculum Promotion*
17:       **for** each batch $\mathcal{B} \subset \mathcal{D}$ **do**
18:        $\tilde{x}_i \leftarrow \mathcal{F}^{-1}(\mathcal{F}(x_i) \odot M_r)$
19:        $\tilde{y}_i \leftarrow f_\theta(\tilde{x}_i)$
20:        $\mathcal{L}_{\text{anti}} \leftarrow \frac{1}{|\mathcal{B}|} \sum_i [L_{BCE}^w(\tilde{y}_i, y_i) + L_{IoU}^w(\tilde{y}_i, y_i)]$
21:        Update $\theta$
22:       **end for**
23:    **end if**
24: **end for**
25: **return** $f_\theta$

---

we exploit temporal statistics of sample-wise performance, termed TSSW, to down-weight problematic samples.

For each sample $i$, we maintain a circular buffer storing its difficulty scores over the past $K$ epochs: $\mathcal{H}_i = \{d_i^{(1)}, \cdots, d_i^{(K)}\}$, computing two temporal statistics:

$$\mu_i = \frac{1}{K} \sum_{k=1}^K d_i^{(k)}, \quad \sigma_i^2 = \frac{1}{K} \sum_{k=1}^K (d_i^{(k)} - \mu_i)^2. \quad (3)$$

These statistics enable CurriSeg to distinguish hard-but-informative samples from noisy or ambiguous ones:

- **High variance** indicates that the model's predictions fluctuate significantly over time, suggesting the sample lies near the decision boundary or contains inherent ambiguity that destabilizes learning.
- **High mean error with low variance** indicates consistent failure, signaling a potential outlier or annotation error that the model cannot resolve.

By first applying min-max normalization to obtain $\tilde{\mu}_i$ and $\tilde{\sigma}_i^2$, we convert the temporal statistics into weights:

$$\omega_i^\mu = 1 - \tilde{\mu}_i, \ \omega_i^\sigma = \exp\left(-\frac{(\tilde{\sigma}_i^2 - \sigma^*)^2}{2\gamma^2}\right), \quad (4)$$

where $\sigma^* = 0.5$ is the optimal variance level. $\gamma = 0.2$

controls tolerance to variance deviation. The final weight is:

$$\omega_i = W_{min}^s + (1 - W_{min}^s) \cdot \omega_i^\mu \cdot \omega_i^\sigma \cdot (1 - \tilde{\mu}_i \cdot (1 - \tilde{\sigma}_i^2)), \quad (5)$$

where $W_{min}^s = 0.1$ provides a weight lower bound and $\cdot$ is product. We denote the last term as $\omega_i^{out}$, *i.e.*, $\omega_i^{out} = 1 - \tilde{\mu}_i \cdot (1 - \tilde{\sigma}_i^2)$. $\omega_i^{out}$ penalizes the "outlier" pattern of high mean error combined with low variance. Hence, CurriSeg emphasizes hard-but-informative samples while suppressing those that are unstable or likely mislabeled.

**Pixel-level uncertainty estimation**. Segmentation exhibits within-sample variation: boundary and low-contrast pixels are harder than homogeneous ones. If treated equally, gradients from uncertain pixels can dominate optimization. We introduce entropy-based pixel weighting (PUE) to attenuate uncertain pixels while preserving informative gradients.

In contrast to prior work that typically uses uncertainty to filter pseudo labels (He et al., 2024a; 2025a;d), we apply uncertainty directly on the predicted mask. The goal is not to hide failure regions from the model, but to prevent them from disproportionately influencing learning.

For a prediction logit $\hat{y}_{h,w}$ at spatial location $(h, w)$, we compute its predicted probability $p_{h,w}$ with Sigmoid $\sigma$: $p_{h,w} = \sigma(\hat{y}_{h,w})$. The normalized prediction entropy is:

$$H_{h,w} = -p_{h,w} \log_2 p_{h,w} - (1 - p_{h,w}) \log_2(1 - p_{h,w}). \quad (6)$$

$H_{h,w} \in [0, 1]$ reaches its maximum when $p_{h,w} = 0.5$ (maximum uncertainty) and minimum when $p_{h,w} \in \{0, 1\}$ (complete certainty). Instead of hard masking, we define a soft weighting matrix $W_i$ whose entry at $(h, w)$ and epoch $t$ is:

$$W_{h,w}(t) = W_{min} + (1 - W_{min}) \cdot (1 - \beta(t) \cdot H_{h,w}), \quad (7)$$

where $W_{min} = 0.1$ sets a non-zero weight floor, and $\beta(t) = (1 - t/T_c)$ is a curriculum-aware coefficient that decays over the curriculum phase of length $T_c$. Early in training, high-entropy pixels receive substantially reduced weights, limiting gradient noise from ambiguous regions. As training progresses and $\beta(t)$ decreases, full supervision is gradually restored, while the weight floor ensures continuous, albeit reduced, gradient flow from all pixels.

**Unified curriculum loss**. Our overall objective follows standard CECS practice (Fan et al., 2020; He et al., 2023a), augmented with the proposed curriculum mechanisms. For a sample $i \in S_t$ at epoch $t$, the loss is defined as:

$$\mathcal{L}_{curr} = \omega_i \cdot [L_{BCE}^w(\hat{y}_i, y_i, W_i) + L_{IoU}^w(\hat{y}_i, y_i, W_i)], \quad (8)$$

where $\omega_i$ is the sample-level weight from temporal statistics. $L_{BCE}^w$ and $L_{IoU}^w$ are the weighted binary cross-entropy loss and weighted intersection-over-union loss with the combined pixel-level weight $W_i$. This joint image- and pixel-level curriculum yields a stable learning schedule.

### 3.2. Anti-Curriculum Promotion

Once the model has reached a stable representation regime, we introduce an Anti-Curriculum Promotion phase to en-

*Table 1.* Results on camouflaged object detection. The suffix "+" indicates that the network is trained under our CurriSeg. $\Delta$: increase.

| Methods | Backbones | CHAMELEON | | | | CAMO | | | | COD10K | | | | NC4K | | | | COD |
|---|---|---|---|---|---|---|---|---|---|---|---|---|---|---|---|---|---|---|
| | | $M \downarrow$ | $F_\beta \uparrow$ | $E_\phi \uparrow$ | $S_\alpha \uparrow$ | $M \downarrow$ | $F_\beta \uparrow$ | $E_\phi \uparrow$ | $S_\alpha \uparrow$ | $M \downarrow$ | $F_\beta \uparrow$ | $E_\phi \uparrow$ | $S_\alpha \uparrow$ | $M \downarrow$ | $F_\beta \uparrow$ | $E_\phi \uparrow$ | $S_\alpha \uparrow$ | $\Delta$ (%) |
| SINet (Fan et al., 2020) | ResNet50 | 0.034 | 0.823 | 0.936 | 0.872 | 0.092 | 0.712 | 0.804 | 0.745 | 0.043 | 0.667 | 0.864 | 0.776 | 0.058 | 0.768 | 0.871 | 0.808 | — |
| MGL-R (Zhai et al., 2021) | ResNet50 | 0.031 | 0.825 | 0.917 | 0.891 | 0.088 | 0.738 | 0.812 | 0.775 | 0.035 | 0.680 | 0.851 | 0.814 | 0.053 | 0.778 | 0.867 | 0.833 | — |
| PreyNet (Zhang et al., 2022a) | ResNet50 | 0.027 | 0.844 | 0.948 | 0.895 | 0.077 | 0.763 | 0.854 | 0.790 | 0.034 | 0.715 | 0.894 | 0.813 | 0.047 | 0.798 | 0.887 | 0.838 | — |
| FEDER (He et al., 2023a) | ResNet50 | 0.028 | 0.850 | 0.944 | 0.892 | 0.070 | 0.775 | 0.870 | 0.802 | 0.032 | 0.715 | 0.892 | 0.810 | 0.046 | 0.808 | 0.900 | 0.842 | — |
| FEDER+ (Ours) | ResNet50 | 0.026 | 0.858 | 0.952 | 0.898 | 0.068 | 0.790 | 0.881 | 0.807 | 0.030 | 0.736 | 0.910 | 0.818 | 0.043 | 0.825 | 0.912 | 0.850 | 2.46 ↑ |
| FSEL (Sun et al., 2024) | ResNet50 | 0.029 | 0.847 | 0.941 | 0.893 | 0.069 | 0.779 | 0.881 | 0.816 | 0.032 | 0.722 | 0.891 | 0.822 | 0.045 | 0.807 | 0.901 | 0.847 | — |
| FSEL+ (Ours) | ResNet50 | 0.028 | 0.856 | 0.949 | 0.898 | **0.067** | 0.792 | **0.889** | **0.819** | 0.030 | 0.742 | 0.909 | 0.831 | 0.042 | 0.823 | 0.919 | 0.855 | 2.22 ↑ |
| RUN (He et al., 2025e) | ResNet50 | 0.027 | 0.855 | 0.952 | 0.895 | 0.070 | 0.781 | 0.868 | 0.806 | 0.030 | 0.747 | 0.903 | 0.827 | 0.042 | 0.824 | 0.908 | 0.851 | — |
| RUN+ (Ours) | ResNet50 | **0.025** | **0.863** | **0.960** | **0.900** | 0.068 | **0.801** | 0.879 | 0.813 | **0.029** | **0.765** | **0.917** | **0.836** | **0.040** | **0.835** | **0.921** | **0.858** | 2.13 ↑ |
| BSA-Net (Zhu et al., 2022) | Res2Net50 | 0.027 | 0.851 | 0.946 | 0.895 | 0.079 | 0.768 | 0.851 | 0.796 | 0.034 | 0.723 | 0.891 | 0.818 | 0.048 | 0.805 | 0.897 | 0.841 | — |
| RUN (He et al., 2025e) | Res2Net50 | 0.024 | 0.879 | 0.956 | 0.907 | 0.066 | 0.815 | 0.905 | 0.843 | 0.028 | 0.764 | 0.914 | 0.849 | 0.041 | 0.830 | 0.917 | 0.859 | — |
| RUN+ (Ours) | Res2Net50 | **0.023** | **0.891** | **0.963** | **0.911** | **0.065** | **0.820** | **0.912** | **0.845** | **0.026** | **0.785** | **0.933** | **0.857** | **0.038** | **0.852** | **0.932** | **0.870** | 2.23 ↑ |
| CamoDiff (Sun et al., 2025) | PVT V2 | 0.022 | 0.868 | 0.952 | 0.908 | 0.042 | 0.853 | 0.936 | 0.878 | 0.019 | 0.815 | 0.943 | 0.883 | 0.028 | 0.858 | 0.942 | 0.895 | — |
| RUN (He et al., 2025e) | PVT V2 | 0.021 | 0.877 | 0.958 | 0.916 | 0.045 | 0.861 | 0.934 | 0.877 | 0.021 | 0.810 | 0.941 | 0.878 | 0.030 | 0.868 | 0.940 | 0.892 | — |
| RUN+ (Ours) | PVT V2 | **0.019** | **0.893** | **0.971** | **0.922** | **0.042** | **0.879** | **0.952** | **0.885** | **0.018** | **0.828** | **0.957** | **0.886** | **0.026** | **0.889** | **0.958** | **0.903** | 3.94 ↑ |

*Table 2.* Training overhead analysis (batch size: 2).

| Metrics | FEDER | FEDER+ | FSEL | FSEL+ | RUN | RUN+ |
|---|---|---|---|---|---|---|
| Training Time (h) | 9.62 | 6.84 | 11.54 | 5.96 | 12.64 | 8.32 |
| GPU Mem (G) | 1.53 | 1.62 | 2.83 | 2.92 | 3.66 | 3.75 |
| Perf. Gain (%) | — | 2.46 ↑ | — | 2.22 ↑ | — | 2.13 ↑ |

*Table 3.* Results on polyp image segmentation.

| Methods | CVC-ColonDB | | | ETIS | | | PIS |
|---|---|---|---|---|---|---|---|
| | mDice ↑ | mIoU ↑ | $S_\alpha \uparrow$ | mDice ↑ | mIoU ↑ | $S_\alpha \uparrow$ | $\Delta$ (%) |
| PolypPVT (Dong et al., 2023) | 0.808 | 0.727 | 0.865 | 0.787 | 0.706 | 0.871 | — |
| CoInNet (Jain et al., 2023) | 0.797 | 0.729 | 0.875 | 0.759 | 0.690 | 0.859 | — |
| LSSNet (Wang et al., 2024) | 0.820 | 0.741 | 0.867 | 0.779 | 0.701 | 0.867 | — |
| MEGANet (Bui et al., 2024) | 0.793 | 0.714 | 0.854 | 0.739 | 0.665 | 0.836 | — |
| MEGANet+ (Ours) | 0.815 | 0.736 | 0.866 | 0.755 | 0.682 | 0.848 | 2.24 ↑ |
| RUN (He et al., 2025e) | 0.822 | 0.742 | 0.880 | 0.788 | 0.709 | 0.878 | — |
| RUN+ (Ours) | **0.845** | **0.763** | **0.893** | **0.806** | **0.720** | **0.890** | 2.05 ↑ |

*Table 4.* Results on transparent object detection.

| Methods | GDD | | | GSD | | | TOD |
|---|---|---|---|---|---|---|---|
| | mIoU ↑ | $F_\beta^{max} \uparrow$ | $M \downarrow$ | mIoU ↑ | $F_\beta^{max} \uparrow$ | $M \downarrow$ | $\Delta$ (%) |
| EBLNet (He et al., 2021) | 0.870 | 0.922 | 0.064 | 0.817 | 0.878 | 0.059 | — |
| IEBAF (Han et al., 2024) | 0.887 | 0.944 | 0.056 | 0.861 | 0.926 | 0.049 | — |
| GhostingNet (Yan et al., 2024) | 0.893 | 0.943 | 0.054 | 0.838 | 0.904 | 0.055 | — |
| RFENet (Fan et al., 2023b) | 0.886 | 0.938 | 0.057 | 0.865 | 0.931 | 0.048 | — |
| RFENet+ (Ours) | 0.893 | 0.952 | 0.052 | 0.875 | 0.939 | 0.045 | 3.22 ↑ |
| RUN (He et al., 2025e) | 0.895 | 0.952 | 0.051 | 0.866 | 0.938 | 0.043 | — |
| RUN+ (Ours) | **0.902** | **0.966** | **0.047** | **0.878** | **0.945** | **0.039** | 3.59 ↑ |

thus enhance the ability of the model in handling hard cases.

Let $\tilde{y}$ denote the predicted mask for $\tilde{x}_i$. The loss function is:

$$\mathcal{L}_{anti} = L_{BCE}^w(\tilde{y}_i, y_i) + L_{IoU}^w(\tilde{y}_i, y_i). \qquad (11)$$

### 3.3. Overall Training Pipeline

Algorithm 1 summarizes the CurriSeg training procedure:

- **Phase 1 (Epochs 1 to $T_c$):** Robust Curriculum Selection builds stable representations through progressive sample introduction, temporal statistics-based weighting, and pixel-level uncertainty estimation.
- **Phase 2 (Epochs $T_c$ to $T$):** Anti-Curriculum Promotion enhances robustness to hard cases through SBFT.

CurriSeg does not modify the model architecture and introduces no extra parameters at inference. The overall training cost remains comparable to standard schedules, while the curriculum phase often accelerates effective convergence by focusing computation on more informative samples.

## 4. Experiments

**Experimental setup**. Our CurriSeg framework is implemented in PyTorch and trained on two NVIDIA RTX 4090 GPUs, using the Adam optimizer with momentum terms parameters $(\beta_1, \beta_2) = (0.9, 0.999)$. We adopt an initial warm-up period of 10 epochs in which all training samples are used without curriculum filtering, allowing the model to acquire a basic representation capacity before difficulty-based selection is applied. Unless specified, the curriculum phase length is set to $T_c = 60$ epochs, followed by an anti-curriculum fine-tuning phase up to $T = 70$ epochs. When

hance its performance on hard samples. This phase counteracts the tendency of the network to exploit superficial texture shortcuts, an inherent bias in dense prediction, and encourages the discovery of complementary, subtler cues.

We propose Spectral-Blindness Fine-Tuning (SBFT), which attenuates high-frequency input components. By suppressing texture information, SBFT forces reliance on low-frequency structural patterns and contextual semantics, promoting extraction of intrinsic, task-relevant signals.

Given the concealed input $x_i$, we apply a low-pass filter in the frequency domain, formulated as:

$$\tilde{x}_i = \mathcal{F}^{-1}(\mathcal{F}(x_i) \odot M_r), \qquad (9)$$

where $\mathcal{F}$ and $\mathcal{F}^{-1}$ denote the 2D Fourier transform and its inverse, and $M_r$ is a circular mask that preserves frequency components within a radius ratio $r$, defined as:

$$M_r(u,v) = \mathbf{1}\left[(u-c_u)^2 + (v-c_v)^2 \leq \frac{r \cdot \min(H,W)}{2}\right], \quad (10)$$

where $\mathbf{1}(\cdot)$ is a indicator function, $H, W$ are the image height and width, $(c_u, c_v)$ is the frequency domain center and $r = 0.95$ controls the cutoff ratio. By attenuating high-frequency texture details, SBFT creates an information bottleneck that reduces reliance on superficial patterns; the model is instead driven to exploit low-frequency boundaries, semantic context, and subtle intensity variations, which tend to be more robust across diverse concealed scenarios and

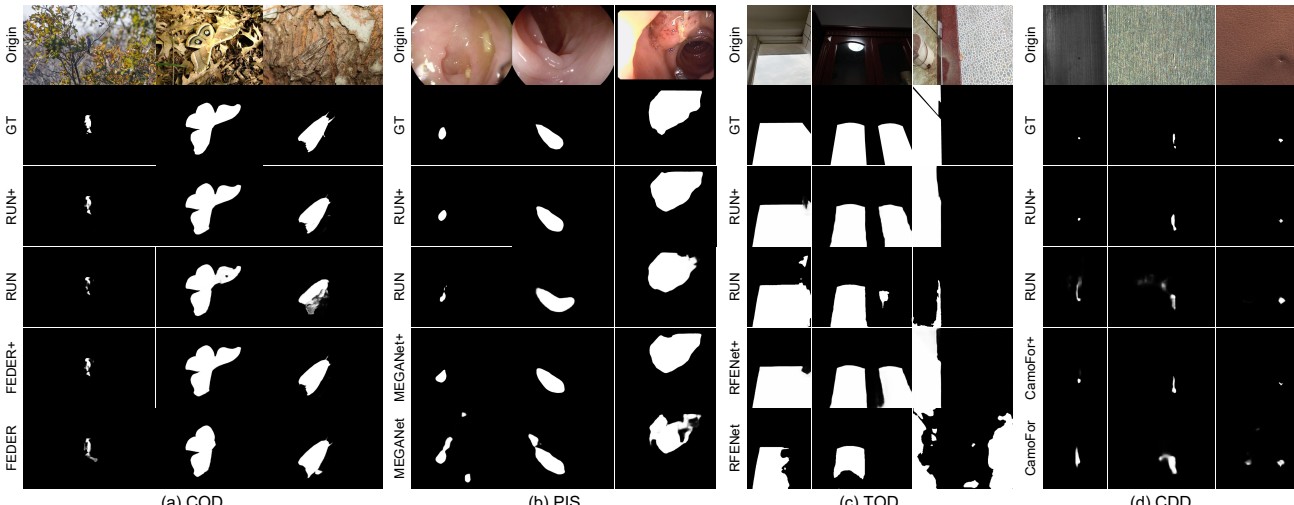

*Figure 3.* Visual comparison on COD, PIS, TOD, and CDD tasks.

*Table 5.* Results on concealed defect detection.

| Methods | $S_\alpha \uparrow$ | $M \downarrow$ | $E_\phi \uparrow$ | $F_\beta \uparrow$ | $F_\beta^{mean} \uparrow$ | $\Delta$ (%) |
|---|---|---|---|---|---|---|
| SINet V2 (Fan et al., 2021a) | 0.551 | 0.102 | 0.567 | 0.223 | 0.248 | — |
| HitNet (Hu et al., 2023) | 0.563 | 0.118 | 0.564 | 0.298 | 0.298 | — |
| OAFormer (Yang et al., 2023) | 0.541 | 0.121 | 0.535 | 0.216 | 0.239 | — |
| CamoFormer (Yin et al., 2024) | 0.589 | 0.100 | 0.588 | 0.330 | 0.329 | — |
| CamoFormer+ (Ours) | **0.599** | 0.091 | 0.603 | **0.345** | **0.346** | 4.59 ↑ |
| RUN (He et al., 2025e) | 0.590 | 0.068 | 0.595 | 0.298 | 0.299 | — |
| RUN+ (Ours) | 0.598 | **0.061** | **0.612** | 0.309 | 0.310 | 4.38 ↑ |

*Table 6.* Breakdown ablations of CurriSeg.

| WCS | PUE | TSSW | SBFT | $M \downarrow$ | $F_\beta \uparrow$ | $E_\phi \uparrow$ | $S_\alpha \uparrow$ |
|---|---|---|---|---|---|---|---|
| × | × | × | × | 0.032 | 0.715 | 0.892 | 0.810 |
| ✓ | × | × | × | 0.035 | 0.697 | 0.870 | 0.801 |
| ✓ | ✓ | × | × | 0.033 | 0.718 | 0.895 | 0.809 |
| ✓ | ✓ | ✓ | × | 0.031 | 0.729 | 0.904 | 0.815 |
| × | × | × | ✓ | 0.031 | 0.723 | 0.902 | 0.813 |
| ✓ | ✓ | ✓ | ✓ | **0.030** | **0.736** | **0.910** | **0.818** |

*Table 7.* Ablation study of the robust curriculum selection component in CurriSeg. We evaluate performance in the COD task on *COD10K*. SPL and TSC are shorts for self-paced learning and teacher-student curriculum. "w/o" indicates without.

| Metrics | WCS | | | | TSSW | | | | | PUE | | CurriSeg |
|---|---|---|---|---|---|---|---|---|---|---|---|---|
| | SPL → WCS | TSC → WCS | $p_1(t) \to p(t)$ | $p_2(t) \to p(t)$ | w/o $\omega_i^\mu$ | w/o $\omega_i^\sigma$ | w/o $\omega_i^{out}$ | $\omega_i^{\sigma 1} \to \omega_i^\sigma$ | $\omega_i^{\sigma 2} \to \omega_i^\sigma$ | w/o $\beta(t)$ | $\beta_1(t) \to \beta(t)$ | (Ours) |
| $M \downarrow$ | **0.030** | **0.030** | 0.031 | 0.031 | 0.031 | 0.031 | 0.030 | 0.031 | 0.031 | 0.031 | 0.030 | **0.030** |
| $F_\beta \uparrow$ | **0.738** | 0.735 | 0.730 | 0.726 | 0.729 | 0.727 | 0.732 | 0.730 | 0.727 | 0.730 | 0.732 | 0.736 |
| $E_\phi \uparrow$ | 0.908 | **0.910** | 0.905 | 0.902 | 0.903 | 0.904 | 0.906 | 0.900 | 0.903 | 0.900 | 0.907 | **0.910** |
| $S_\alpha \uparrow$ | **0.820** | 0.817 | 0.815 | 0.812 | 0.814 | 0.810 | 0.810 | 0.808 | 0.809 | 0.814 | 0.815 | 0.818 |

*Table 8.* Ablation study of anti-curriculum promotion in CurriSeg. TAL and AA are texture-aware loss and aggressive augmentation.

| Metrics | Hard-only SBFT | Random SBFT | Square filter | Progressive filter | Blur → SBFT | Noise → SBFT | TAL → SBFT | AA → SBFT | Reverse CurriSeg | CurriSeg |
|---|---|---|---|---|---|---|---|---|---|---|
| $M \downarrow$ | 0.031 | 0.031 | 0.031 | **0.030** | 0.033 | 0.032 | 0.032 | 0.033 | 0.052 | **0.030** |
| $F_\beta \uparrow$ | 0.730 | 0.728 | 0.729 | 0.734 | 0.710 | 0.719 | 0.725 | 0.718 | 0.632 | **0.736** |
| $E_\phi \uparrow$ | 0.899 | 0.902 | 0.905 | **0.910** | 0.875 | 0.866 | 0.897 | 0.868 | 0.787 | **0.910** |
| $S_\alpha \uparrow$ | 0.813 | 0.812 | 0.814 | 0.816 | 0.804 | 0.806 | 0.808 | 0.807 | 0.742 | **0.818** |

integrating CurriSeg with existing CECS backbones, all input images are resized to $352 \times 352$ during both training and testing, and all other hyperparameters follow the original configurations of the corresponding backbone methods.

### 4.1. Comparative Evaluation

We conduct extensive experiments across CECS tasks, with datasets and metrics detailed in Sec. A.1 in the appendix. All results follow standardized evaluation protocols.

**Camouflaged object detection**. As shown in Table 1, integrating our CurriSeg into cutting-edge methods (denoted by "+") consistently improves performance across ResNet50 (He et al., 2016), Res2Net50 (Gao et al., 2019), and PVT V2 (Wang et al., 2022). Moreover, the qualitative results in Fig. 3 demonstrate that models optimized under our CurriSeg yield more accurate segmentation, even in

highly challenging scenarios. Together, these results substantiate the effectiveness of our dual-phase curriculum strategy. Beyond performance gains, CurriSeg also improves training efficiency. As reported in Table 2, training time is reduced by 28.9% -48.4% across backbones, since curriculum selection concentrates computation on informative samples rather than all data. GPU memory overhead is negligible ($\leq$0.1G), since no extra parameters are introduced. Hence, our CurriSeg is more efficient than the standard manner.

**Polyp image segmentation**. Experiments are conducted on the PIS task using *CVC-ColonDB* and *ETIS*. In line with RUN (He et al., 2025e), PVT V2 is employed as the backbone. As evidenced by the quantitative results in Table 3 and the visualizations in Fig. 3, models trained under our CurriSeg consistently outperform their original versions.

**Transparent object detection**. As reported in Table 4 and Fig. 3, CurriSeg integration yields consistent gains over

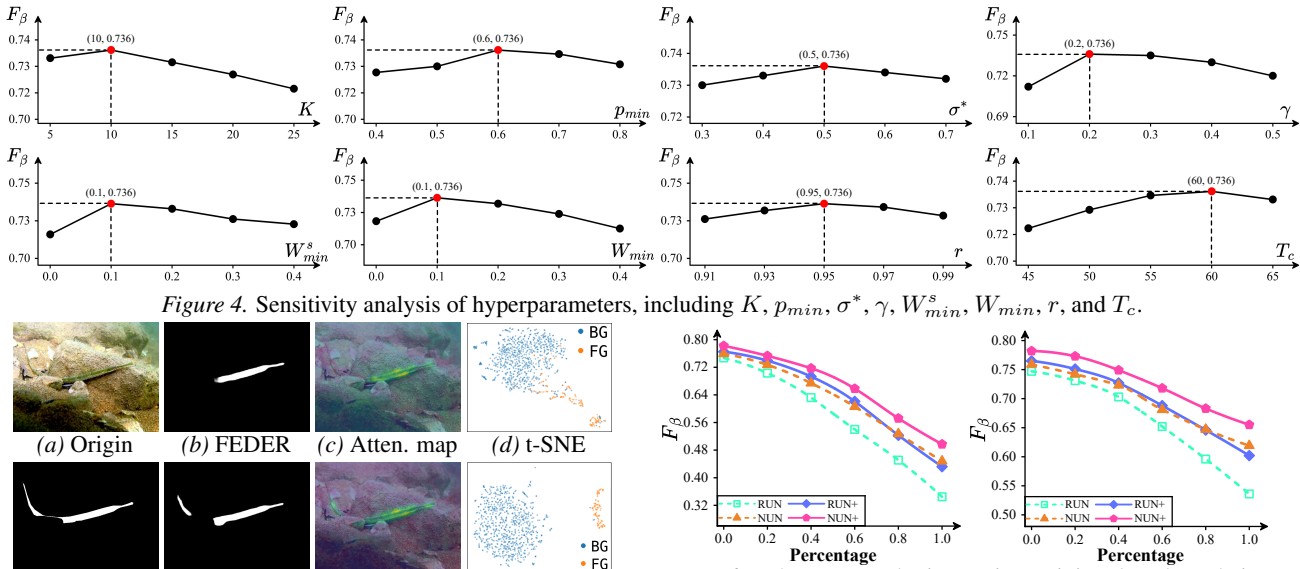

*Figure 4.* Sensitivity analysis of hyperparameters, including $K$, $p_{min}$, $\sigma^*$, $\gamma$, $W^s_{min}$, $W_{min}$, $r$, and $T_c$.

|  |  |  |  |
|---|---|---|---|
| *(a)* Origin | *(b)* FEDER | *(c)* Atten. map | *(d)* t-SNE |
| *(e)* GT | *(f)* FEDER+ | *(g)* Atten. map | *(h)* t-SNE |

*Figure 5.* Visualization of attention map and t-SNE of foreground (orange) and background (blue) features. (b)-(d) and (f)-(h) correspond to the prediction, attention map, and t-sne of FEDER trained under the standard setting and our CurriSeg framework.

*Figure 6.* Robustness under increasing training data degradation. **Left**: High-frequency information loss (blur, low-light, haze). **Right**: High-frequency interference (noise). CurriSeg maintains superior performance across all degradation ratios, indicating the potential of our framework in real-world degradation scenarios.

*Table 9.* Generalization of CurriSeg on other settings, including weak supervision with scribble on *COD10K* (WS-SAM (He et al., 2024a) and SEE (He et al., 2025a)), semi-supervision with 1/16 labeled data on *COD10K* (CoSOD (Chakraborty et al., 2024) and SEE), multi-modality with depth on *COD10K* (DSAM (Yu et al., 2024) and MultiCOS (Fang et al., 2025)), and video segmentation on *CAD* (Cheng et al., 2022b) (STL-Net (Cheng et al., 2022b) and ZoomNext (Pang et al., 2024)). Metrics are reported according to each setting.

*(a)* Weak supervision.

| Methods | $M \downarrow$ | $F_\beta \uparrow$ | $E_\phi \uparrow$ | $S_\alpha \uparrow$ |
|---|---|---|---|---|
| WS-SAM | 0.038 | 0.719 | 0.878 | 0.803 |
| WS-SAM+ | 0.035 | 0.733 | 0.886 | 0.813 |
| SEE | 0.036 | 0.729 | 0.883 | 0.807 |
| SEE+ | **0.033** | **0.738** | **0.893** | **0.816** |

*(b)* Semi-supervision.

| Methods | $M \downarrow$ | $F_\beta \uparrow$ | $E_\phi \uparrow$ | $S_\alpha \uparrow$ |
|---|---|---|---|---|
| CoSOD | 0.055 | 0.650 | 0.795 | 0.740 |
| CoSOD+ | 0.051 | 0.672 | 0.810 | 0.749 |
| SEE | 0.046 | 0.679 | 0.803 | 0.745 |
| SEE+ | **0.043** | **0.695** | **0.817** | **0.757** |

*(c)* Multi-modality.

| Methods | $M \downarrow$ | $F_\beta^{max} \uparrow$ | $E_\phi^{max} \uparrow$ | $S_\alpha \uparrow$ |
|---|---|---|---|---|
| DSAM | 0.033 | 0.807 | 0.931 | 0.846 |
| DSAM+ | 0.030 | 0.821 | 0.942 | 0.853 |
| MultiCOS | 0.020 | 0.950 | 0.946 | 0.880 |
| MultiCOS+ | **0.018** | **0.958** | **0.953** | **0.883** |

*(d)* Video segmentation.

| Methods | $M \downarrow$ | $F_\beta^w \uparrow$ | $E_\phi^{max} \uparrow$ | $S_\alpha \uparrow$ |
|---|---|---|---|---|
| STL-Net | 0.030 | 0.481 | 0.845 | 0.696 |
| STL-Net+ | 0.026 | 0.505 | 0.861 | 0.706 |
| ZoomNext | 0.020 | 0.593 | 0.865 | 0.757 |
| ZoomNext+ | **0.018** | **0.611** | **0.873** | **0.761** |

*Figure 7.* Performance ($F_\beta$) vs. training epoch on COD10K. Phase 1: 0-$T_c$ and Phase 2: $T_c$-$T$. The initial 10 epochs are used for warm-up, where FEDER shares the same training details with FEDER+, and thus be omitted for clarification.

existing methods, highlighting its utility as a general performance enhancer for perception in autonomous driving.

**Concealed defect detection**. We validate the generalizability of models trained with CurriSeg on CDD. Models pre-trained on COD are directly transferred to the *CDS2K* dataset for defect segmentation, following RUN (He et al., 2025e). As shown in Table 5 and Fig. 3, models optimized under CurriSeg consistently achieve superior performance,

demonstrating enhanced cross-domain generalization.

## 4.2. Ablation Study

We conduct ablations on *COD10K* using FEDER, a cutting-edge network that integrates frequency guidance and attention mechanisms, as the baseline in Secs. 4.2 and 4.3.

**Effect of robust curriculum selection**. As shown in Table 6, robust curriculum selection (WCS+PUE+TSSW) stabilizes training and suppresses ambiguous supervision. In Table 7, replacing WCS with self-paced learning (Kumar et al., 2010) or teacher-student curricula (Matiisen et al., 2019) yields comparable results but adds complexity. Altering the warm-up schedule from $P(t)$ to $p_1(t) = p_{min} + (1 - p_{min}) \cdot (\frac{t-1}{T_c - 1})^2$ or $p_2(t) = p_{min} + (1 - p_{min}) \cdot (\frac{t-1}{T_c - 1})^{0.5}$ degrades performance, suggesting $p(t)$ offers a better stability-coverage trade-off. Within TSSW, dropping any component ($\omega_i^\mu$, $\omega_i^\sigma$, and $\omega_i^{out}$) or replacing $\omega_i^\sigma$ with $\omega_i^{\sigma 1} = \max(0, 1 - \frac{|\tilde{\sigma}_i^2 - \sigma^*|}{\gamma})$ and $\omega_i^{\sigma 2} = 1 - (\frac{\tilde{\sigma}_i^2 - \sigma^*}{\gamma})^2$ leads to noticeable drops. Finally, removing the entropy-based pixel reweighting $\beta(t)$ or using exponential decay $\beta_1(t) = \exp(-\frac{t}{T_c})$ also hurts.

**Effect of anti-curriculum promotion**. As shown in Ta-

*Table 10.* Generalization of CurriSeg on other dense prediction tasks with cutting-edge methods, including semantic segmentation on *ADE20K* (Zhou et al., 2017) (PEM (Cavagnero et al., 2024) and CGRSeg (Ni et al., 2024)), instance segmentation on *COCO* (Lin et al., 2014) (M2Form and FasIns are short for Mask2Former (Cheng et al., 2022a) and FastInst (He et al., 2023b)), infrared small target detection on *IRSTD-1k* (Zhang et al., 2022b) (ISNet (Zhang et al., 2022b) and IRSAM (Zhang et al., 2024)), and shadow detection on *SBU* (Vicente et al., 2016) (SARA (Sun et al., 2023) and spider (Zhao et al., 2024)), with BER (Vicente et al., 2015) used for evaluation.

| (a) Semantic segmentation. | | | | (b) Instance segmentation. | | | | (c) Infrared small target detection. | | | | (d) Shadow detection. | | | |
| --- | --- | --- | --- | --- | --- | --- | --- | --- | --- | --- | --- | --- | --- | --- | --- |
| Metric | PEM | PEM+ | CGRSeg | CGRSeg+ | Metric | M2Form | M2Form+ | FasIns | FasIns+ | Metric | ISNet | ISNet+ | IRSAM | IRSAM+ | Metric | SARA | SARA+ | Spider | Spider+ |
| IoU ↑ | 45.0 | 45.9 | 45.5 | **46.8** | AP ↑ | 38.0 | 38.6 | 38.6 | **39.1** | IoU ↑ | 68.77 | 70.62 | 73.69 | **75.52** | BER ↓ | 0.043 | 0.040 | 0.040 | **0.038** |

*Table 11.* Compatibility of our CurriSeg with advanced architectures (attention-based FSEL (Sun et al., 2024), multiscale-based ZoomNet (Pang et al., 2022), and uncertainty-based UGTR (Yang et al., 2021)) and foundation models (SAM-adapter (SAM-a) (Chen et al., 2023), SAM2-adapter (SAM2-a) (Chen et al., 2024), and SAM3-adapter (SAM3-a) (Chen et al., 2025)) on *COD10K*. In (b)-(d), the suffix "*" means modifying the general CurriSeg framework with simple, architecture-specific changes. The further performance improvement indicates the potential of our CurriSeg in facilitating the community.

| (a) | (b) Attention. | | | (c) Multiscale. | | | (d) Uncertainty. | | | (e) Foundation model adapters. | | | | | |
| --- | --- | --- | --- | --- | --- | --- | --- | --- | --- | --- | --- | --- | --- | --- | --- |
| Metr. | FSEL | FSEL+ | FSEL* | ZoomNet | ZoomNet+ | ZoomNet* | UGTR | UGTR+ | UGTR* | SAM-a | SAM-a+ | SAM2-a | SAM2-a+ | SAM3-a | SAM3-a+ |
| $M \downarrow$ | 0.032 | 0.030 | **0.029** | 0.029 | 0.028 | **0.027** | 0.036 | 0.034 | **0.032** | 0.025 | 0.023 | 0.018 | 0.016 | 0.015 | **0.014** |
| $F_\beta \uparrow$ | 0.722 | 0.742 | **0.749** | 0.740 | 0.753 | **0.758** | 0.670 | 0.688 | **0.706** | 0.800 | 0.822 | 0.848 | 0.869 | 0.883 | **0.892** |
| $E_\phi \uparrow$ | 0.891 | 0.909 | **0.916** | 0.888 | 0.898 | **0.907** | 0.852 | 0.865 | **0.889** | 0.918 | 0.935 | 0.950 | 0.962 | 0.965 | **0.968** |
| $S_\alpha \uparrow$ | 0.822 | 0.831 | **0.834** | 0.838 | 0.842 | **0.845** | 0.817 | 0.825 | **0.830** | 0.883 | 0.888 | 0.899 | 0.903 | 0.927 | **0.930** |

| (f) Origin | (g) Degradation | (h) GT | (i) Initial | (j) Mid | (k) Final | (l) + Anti-Curri | (m) + CurriSeg |

*Figure 8.* Robustness analysis under high-frequency loss (Row 1: haze) and interference (Row 2: noise). While converged FEDER (f) overfit fragile texture shortcuts, CurriSeg preserves structural integrity by balancing low-frequency context with task-relevant details.

ble 8, applying SBFT only to hard samples (top 50%) or to a random 50% subset falls below CurriSeg. Replacing our circular low-pass mask with square or progressive filters worsens results. with Gaussian blur, additive noise, texture-aware loss, or aggressive augmentation degrades metrics, confirming the benefit of principled spectral design. Reversing the phase order causes dramatic performance drops.

**Hyper-parameter sensitivity**. Fig. 4 shows CurriSeg's sensitivity to key hyper-parameters. Optimal settings are: $K = 10$, $p_{min} = 0.6$ (WCS); $\sigma^* = 0.5$, $\gamma = 0.2$, $W^s_{min} = 0.1$ (TSSW); $W_{min} = 0.1$ (PUE); $r = 0.95$, $T_c = 60$ (ACP). The framework is robust within moderate ranges.

### 4.3. Further Analysis and Applications

**Feature space analysis**. As shown in Fig. 5, we visualize attention maps and feature distributions via t-SNE (Maaten & Hinton, 2008). The baseline exhibits substantial cluster overlap, while CurriSeg training yields improved intra-class compactness and inter-class separation, confirming that our CurriSeg induces more discriminative representations.

**Broad generalization across settings and tasks**. We evaluate CurriSeg under weak/semi-supervision, multi-modal learning, and video segmentation (Table 9), as well as semantic/instance segmentation, infrared target detection, and shadow detection (Table 10). Consistent improvements

demonstrate that our paradigm provides a general training principle for dense prediction, highlighting our potential.

**Performance dynamics across epochs**. Fig. 7 shows that CurriSeg consistently surpasses baselines throughout training (averaged over five runs), with faster convergence and reduced variance indicating more stable optimization.

**Robustness under image degradation**. We evaluate robustness by progressively introducing degraded samples (from NUN (He et al., 2025f)) with ratios from 0% to 100%. As shown in Fig. 6, CurriSeg outperforms baselines, with the gap widening at higher degradation levels, demonstrating enhanced reliance on low-frequency structural cues.

**Compatibility with advanced architectures**. CurriSeg is architecture-agnostic. Table 11 shows consistent gains on attention-based (FSEL), multi-scale (ZoomNet), uncertainty-aware (UGTR), and foundation model adapters (SAM series). "+": CurriSeg; "*": simple and specific adaptations (Sec. A.2.1), highlighting broad applicability.

## 5. Discussion

CurriSeg offers a frequency-aware perspective extending beyond CECS. **(i) Spectral learning dynamics**: The "stabilize-then-perturb" trajectory first anchors the model on reliable representations through noise-resistant curriculum.

**(ii) Mitigating high-frequency dependency**: SBFT's effectiveness suggests standard training drifts toward texture shortcuts. As shown in Fig. 8, fully converged baselines often overfit fragile high-frequency cues, leading to collapse under spectral degradations. **(iii) Robustness via information bottlenecks**: By attenuating high-frequency components, CurriSeg enforces reliance on low-frequency boundaries and contextual semantics—a powerful, architecture-agnostic complement to frequency-aware modules.

## 6. Conclusions

We proposed CurriSeg, which follows a "stabilize-then-perturb" process: robust curriculum selection uses temporal loss statistics and pixel-level uncertainty to construct a stable, noise-resistant learning schedule, and an anti-curriculum phase applies SBFT to attenuate high-frequency texture cues. This encourages reliance on structural and contextual information and reduces shortcut behavior. Experiments show consistent performance and robustness gains without extra parameters or training overhead, highlighting our importance for entangled and dense visual scenes.

## Acknowledgment

This work was supported in part by the Foundation Fighting Blindness (BR-CL-0621-0812-DUKE and PPA-1224-0890-DUKE) and Research to Prevent Blindness Unrestricted Grant to Duke University. The first two authors contribute equally to this paper.

## Impact Statement

This work advances segmentation for concealed and context-entangled objects, with potential benefits in medical image analysis and autonomous perception. As with other segmentation systems, deployment in safety-critical settings such as clinical diagnosis or driving requires careful validation, since segmentation errors may carry real-world consequences. We do not foresee additional ethical concerns beyond those common to general dense-prediction research.

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

# A. Experiment

## A.1. Datasets and Metrics

**Camouflaged object detection**. Following the protocol of SINet (Fan et al., 2020), we conduct experiments on four datasets: *CHAMELEON* (Skurowski et al., 2018), *CAMO* (Le et al., 2019), *COD10K* (Fan et al., 2021a), and *NC4K* (Lv et al., 2021). The *CHAMELEON* dataset contains 76 images, while *CAMO* comprises 1,250 images from 8 categories. *COD10K* includes 5,066 images organized into 10 super-classes, and *NC4K* is the largest test set, with 4,121 images. For training, we use 1,000 images from *CAMO* and 3,040 images from *COD10K*; the remaining images from these two datasets, together with all images from *CHAMELEON* and *NC4K*, are reserved for testing. We report four widely adopted metrics: mean absolute error ($M$), adaptive F-measure ($F_\beta$) (Margolin et al., 2014), mean E-measure ($E_\phi$) (Fan et al., 2021b), and structure measure ($S_\alpha$) (Fan et al., 2017). Lower $M$ and higher $F_\beta$, $E_\phi$, and $S_\alpha$ indicate superior performance.

**Polyp image segmentation**. For polyp segmentation, we evaluate on two public benchmarks: *CVC-ColonDB* (Tajbakhsh et al., 2015) and *ETIS* (Silva et al., 2014). The training protocol follows the setting of LSSNet (Wang et al., 2024). Quantitative performance is assessed using three commonly employed metrics: mean Dice (mDice), mean Intersection over Union (mIoU), and structure measure ($S_\alpha$), where larger values indicate better performance. To ensure a fair comparison with recent state-of-the-art approaches, which predominantly adopt transformer-based encoders, we use PVT V2 as the encoder backbone.

**Transparent object detection**. For transparent object detection, we also adopt PVT V2 as the default backbone to enable a fair comparison with existing methods. Experiments are conducted on two datasets, *GDD* (Mei et al., 2020) and *GSD* (Lin & He, 2021). The training set consists of 2,980 images from *GDD* and 3,202 images from *GSD*, with the remaining images used exclusively for inference. Following GDNet-B (Mei et al., 2023), we evaluate performance using mIoU and the maximum F-measure ($F_\beta^{max}$), together with mean absolute error ($M$). Better performance corresponds to lower $M$ and higher mIoU and $F_\beta^{max}$.

**Concealed defect detection**. For concealed defect detection, we employ PVT V2 as the default backbone and assess the generalization capability of the proposed RUN framework. Specifically, we directly apply the model trained on the COD task to segment concealed defects on the *CDS2K* dataset (Fan et al., 2023a). Five evaluation metrics are reported; for all metrics except MAE, larger values indicate better performance, whereas for MAE lower values are preferable.

## A.2. Further Analysis

### A.2.1. IMPLEMENTATION DETAILS OF "COMPATIBILITY WITH ADVANCED ARCHITECTURES"

"*" in Table 11 indicates simple architecture-specific adaptations, bringing better performance. The implementation details are presented as follows:

**Curri\* (Attention):** During SBFT, we randomly mask a frequency band within radius range $[r_{low}, r_{high}]$ where $r_{high} - r_{low} = 0.05$, encouraging the attention mechanism to discover scale-invariant patterns.

**Curri\* (Multi-scale):** We apply multi-scale SBFT with different cutoff radii $r$ per scale, and tighten the uncertainty thresholds $W_{min}$ and $W_{min}^s$ in RCS, as multi-scale features inherently reduce prediction instability.

**Curri\* (Uncertainty):** We combine the feature-level uncertainty proposed by UGTR (Yang et al., 2021) from temporal statistics with the existing pixel-level uncertainty, jointly suppressing unreliable gradients at both granularities.

The consistent improvements demonstrate that CurriSeg complements rather than conflicts with existing design principles, offering a promising direction for enhancing diverse segmentation frameworks.

