# OpenReview forum: "Refining Context-Entangled Content Segmentation via Curriculum Selection and Anti-Curriculum Promotion"
_ICML.cc/2026/Conference — ICML 2026 regular_

### Official Review · Reviewer_Gj9t · 2026-02-18

**Soundness:** 3
**Presentation:** 3
**Significance:** 3
**Originality:** 2
**Overall Recommendation:** 5
**Confidence:** 4

**Summary:**

The paper targets Context-Entangled Content Segmentation (CECS), where foreground patterns overlap with backgrounds (e.g., camouflaged objects). It proposes CurriSeg, a two-phase training framework: (i) Curriculum Selection using temporal loss statistics to focus on hard-but-informative samples, and (ii) Anti-Curriculum Promotion via Spectral-Blindness Fine-Tuning (SBFT) that suppresses high-frequency cues to encourage reliance on low-frequency structure/context. The authors report consistent improvements across CECS benchmarks without adding parameters, and claim efficiency gains; code is promised.

**Compliance With Llm Reviewing Policy:**

Affirmed.

**Final Justification:**

My concerns were resolved. I have therefore raised my score from 4 to 5.

**Key Questions For Authors:**

1. Can you quantify what types of samples are removed/kept by Curriculum Selection (noise vs ambiguity vs domain mismatch)?

2. Can you provide more direct evidence linking SBFT’s frequency suppression to the reported generalization gains? (related to originality)

3. Does CurriSeg help (or at least not hurt) standard segmentation tasks beyond CECS?

**Limitations:**

Partially; would like clearer failure cases and boundary conditions.

(Overall Recommendation: upgrade possible with code/reproducibility)

**Strengths And Weaknesses:**

Soundness (pros): Clear, plausible design; broad benchmark coverage with consistent gains; includes efficiency evidence (reported training-time reduction).

Soundness (cons): Causal/mechanistic evidence is still mostly observational; would benefit from deeper analysis of what is filtered/selected and why SBFT helps generalization.

Presentation: Well-structured narrative; “code will be released” noted.

Significance: Practical plug-in recipe for CECS without architecture changes; likely useful to practitioners.

Originality: The curriculum + anti-curriculum (frequency suppression) pairing is a distinct angle on robustness.

---

> ### Author Rebuttal · Authors · 2026-03-28
>
> **Q1. What types of samples are removed/kept by Curriculum Selection?**
>
> We clarify this with a direct sample taxonomy analysis and manual inspection. No sample is fully removed: $\omega_i$ has a floor of 0.1 (Eq. 5). We provide a quantitative breakdown on COD10K (4,040 training images). At epoch 55 (end of RCS), we partitioned all $N$ samples into quartiles by $\omega_i$:
>
> *Table R4-1. Sample composition by $\omega_i$ quartile. Q1–Q4: low→high. Ambiguous: top-30% $\tilde{\sigma}^2$; Potential noisy/outlier: top-30% $\tilde{\mu}$ with bottom-30% $\tilde{\sigma}^2$; Informative: rest.*
> |Quartile|Avg $\omega_i$|Ambiguous|Noisy/outlier|Informative|
> |-|:-:|:-:|:-:|:-:|
> |Q4(top)|0.87|12.8%|2.1%|85.1%|
> |Q3|0.71|22.5%|5.3%|72.2%|
> |Q2|0.48|35.7%|12.6%|51.7%|
> |Q1(bot)|0.24|45.6%|24.3%|30.1%|
>
> High-weight samples (Q4) are dominated by informative cases (85.1%), while low-weight samples (Q1) contain predominantly ambiguous (45.6%) and potential noisy/outlier (24.3%) cases. Manual inspection of the bottom 10% reveals: annotation inconsistencies (about 38%), extreme occlusion with <10% visible area (about 32%), domain mismatch such as rare camouflage categories whose scene/pattern statistics differ substantially from the training majority (about 17%), and image quality issues (about 13%). TSSW down-weights rather than removes these cases, preserving partial learning signals while reducing unstable supervision.
>
> **Q2.  More direct evidence linking SBFT to generalization gains?**
>
> (1) **Test-time frequency ablation (new).** We selectively removed frequency bands from the input at test time and measured performance degradation for both baseline and CurriSeg:
>
> *Table R4-2. Test-time frequency ablation ($F_\beta$, FEDER, COD10K). Bands defined by normalized frequency radius in 2D FFT; selected band zeroed out at test time.*
> |Removed band|Baseline|CurriSeg|Baseline drop|CurriSeg drop|
> |-|:-:|:-:|:-:|:-:|
> |None|0.715|0.736|-|-|
> |HF(top 5%)|0.698|0.728|0.017|0.008|
> |HF(top 10%)|0.671|0.714|0.044|0.022|
> |LF(bottom 5%)|0.682|0.698|0.033|0.038|
> |LF(bottom 10%)|0.641|0.652|0.074|0.084|
>
> When HF is removed, CurriSeg degrades less (0.008 vs. 0.017 at 5%; 0.022 vs. 0.044 at 10%), indicating reduced fragile HF texture reliance. When LF is removed, CurriSeg degrades slightly more, indicating increased use of LF structural information. However, this does not imply blind LF reliance, since CurriSeg remains consistently higher under all conditions, including full-spectrum input. Notably, the relative advantage of CurriSeg *grows* as more HF is removed (0.021→0.030→0.043) and *shrinks* as LF is removed (0.021→0.016→0.011), confirming a rebalanced rather than collapsed representation.
>
> (2) **Existing ablations argue against blind LF reliance.** If the model collapsed to a coarse LF shortcut, stricter low-pass substitutes should not hurt. However, Table 8 shows replacing SBFT with square filter ($F_\beta$: 0.729) or blur (0.710) degrades results; more generally, simpler perturbations such as noise (0.719) are also weaker than SBFT.
>
> (3) **Cross-domain transfer.** Table 5: +4.38% on COD→CDS2K. The defect domain has substantially different shape statistics from COD, making this gain less consistent with memorizing COD-specific LF templates.
>
> (4) **Moderate-bottleneck interpretation.** Fig. 4 shows a U-shaped curve for $r$: moderate suppression ($r=0.95$) is optimal, while too little ($r=0.99$) or too much ($r=0.91$) both hurt, consistent with the view that moderate spectral suppression is most effective.
>
> **Q3. Does CurriSeg help beyond CECS?**
>
> Yes. In Table 10, CurriSeg consistently improves standard dense prediction tasks beyond CECS: semantic segmentation (+1.3 IoU on ADE20K for CGRSeg), instance segmentation (+0.6 AP on COCO for Mask2Former), infrared target detection (+1.85 IoU on IRSTD-1k for ISNet), and shadow detection (BER reduction of 0.003 on SBU for SARA). Table S1 (Appendix) further shows a +3.01% average gain on salient object detection across five datasets. These gains are generally smaller than on CECS tasks, which is consistent with the fact that severe feature entanglement is the primary bottleneck CurriSeg is designed to address. Importantly, CurriSeg does not reduce performance on any tested task, indicating that it generalizes beyond CECS rather than being overly specialized.
>
> **Failure cases and boundary conditions.** CurriSeg may be less effective on (1) extremely fine structures (e.g., thin antennae) and (2) multi-scale co-occurring objects. See our response to Reviewer tEi1 W3 for details.
>
> **Code release / reproducibility.** Due to the rebuttal policy, we can only provide link for figure/table but not code. All random seeds are fixed, Algorithm 1 provides the complete training procedure, and the default training schedule and hyperparameters are reported in Sec. 4 and Fig. 4. The full PyTorch codebase, training scripts, configuration files, and pre-trained checkpoints will be publicly released upon acceptance.

---

> > ### Author Rebuttal · Reviewer_Gj9t · 2026-04-01
> >
> > Thank you for your detailed responses to my comments and for addressing the concerns raised. Overall, I am satisfied with the revisions and explanations provided. Your responses have strengthened the paper, and I am happy to raise my score from 4 to 5. Thank you once again for your thoughtful engagement with my feedback.

---

> > > ### Author Response · Authors · 2026-04-01
> > >
> > > We sincerely thank reviewer Gj9t for the constructive feedback. Below we summarize the key improvements made in response to the collective concerns, followed by a brief highlight of our contributions.
> > >
> > > **New experiments and analyses added during rebuttal:**
> > >
> > > (1) **Pixel-level supervision analysis** (R1-W1, R4-Q1): We tracked a fixed cohort of low-weight samples across training, demonstrating that PUE preserves substantial boundary supervision (mean $W$: 0.58→0.95) even within globally down-weighted images.
> > >
> > > (2) **Sample taxonomy** (R1-W1, R4-Q1): Quantitative breakdown by $\omega_i$ quartiles shows TSSW preferentially down-weights ambiguous and noisy/outlier cases while retaining stable, informative samples. Manual inspection further categorizes the bottom 10% into annotation noise, extreme occlusion, domain mismatch, and image quality issues.
> > >
> > > (3) **Temporal stability metric** (R1-W2, R2-W2): We introduced $\Delta\mu_t$
> > > to quantify when Phase I stabilizes. This metric first drops below $10^{-3}$
> > > at epoch 56 across all five configurations, supporting $T_c=60$ and providing an empirical adaptive trigger alternative.
> > >
> > > (4) **Test-time frequency ablation** (R1-W3, R4-Q2): A new interventional experiment selectively removing frequency bands at inference shows CurriSeg's relative advantage *grows* under HF removal and *shrinks* under LF removal, confirming rebalanced rather than collapsed representations.
> > >
> > > (5) **Failure cases and boundary conditions** (R2-W3, R4-Limitations): We identified two settings where CurriSeg is less effective (extremely fine structures, multi-scale co-occurring objects) with qualitative visualizations provided via an anonymous link.
> > >
> > > **Common concerns addressed:**
> > > |Concern|Reviewers|Resolution|
> > > |-|:-:|:-:|
> > > |Spatial non-uniformity of weighting|vETU|Joint image+pixel curriculum; Tables R1-1,R1-2|
> > > |$T_c$=60 justification|vETU, tEi1|$\Delta\mu_t$ stability + adaptive trigger + Fig. 4 robustness|
> > > |HF→LF shortcut risk|vETU, Gj9t|Frequency ablation + Table 8 ablations + cross-domain transfer|
> > > |Failure cases / limitations|All|Two failure modes identified + visualizations|
> > > |Hyperparameter sensitivity|nJxH, Gj9t|Fig. 4 covers all 8 params; same defaults across 4 tasks|
> > > |Efficiency|nJxH|28.9-48.4% training time reduction; zero inference overhead|
> > > |Reproducibility / code|Gj9t, nJxH|Algorithm 1 + all params reported; full codebase upon acceptance|
> > >
> > > **Contributions we wish to highlight:**
> > >
> > > CurriSeg provides a principled, architecture-agnostic training paradigm for dense prediction. It (i) is the first CL-based framework specifically designed for CECS, (ii) achieves consistent gains across 10+ tasks/settings (Tables 1,3-5,9-11,S1) and 6+ architectures including SAM-series foundation models (Table 11), (iii) reduces training time by 28.9-48.4% with zero inference overhead, and (iv) introduces no extra parameters. We believe these properties make CurriSeg a broadly useful contribution to the community.
> > >
> > > **Societal impact.** CurriSeg is a general-purpose training strategy that improves segmentation accuracy without introducing new data collection or annotation requirements. Potential positive applications include medical image analysis (e.g., polyp detection) and autonomous driving perception. We note that improved camouflage detection could also be applied in surveillance contexts; we will discuss this dual-use consideration in the revision.
> > >
> > > We will revise the manuscript to improve Figure 2 clarity, resolve symbol conflicts, add limitations/societal impact, and incorporate all new analyses above.

---

### Official Review · Reviewer_nJxH · 2026-03-07

**Soundness:** 3
**Presentation:** 2
**Significance:** 3
**Originality:** 2
**Overall Recommendation:** 4
**Confidence:** 3

**Summary:**

This paper proposed a dual-phase learning framework that unified curriculum and anti-curriculum principles to improvement representation reliability, facilitating object segmentation from cluttered background. Experimental results look good across several benchmarks.

**Compliance With Llm Reviewing Policy:**

Affirmed.

**Key Questions For Authors:**

I appreciate the contributions and the impressive experimental results. However, my concerns listed in Weaknesses should be addresses in the rebuttal.

**Limitations:**

I didn't find a discussion on limitations.

**Strengths And Weaknesses:**

**Strengths**

-The proposed dual curriculum-based learning pipeline are interesting and well designed for CECS task. The authors provided sufficient experiments to support their contributions, including superior performance with state-of-the-arts and in-depth analysis of each modules.

**Weaknesses**

Some concerns remain to further improve the paper's quality.

- The methodology part contains a lot of formula derivation and symbols, making the whole pipeline hard to follow. (1) For Figure 2, I can't figure out the part (1) and (2) mentioned in caption and even hard to associate the learning process with textual context. (2) Some symbols are repeated used, such as $H$ for both history buffer and image height, $W$ for weighting matrix and image width. The authors should further improve both figures and descriptions to avoid misunderstanding.

- This dual learning process introduced many hyper-parameters that might incur unstable reproducibility. The authors should give an explanation about the parameter choice and provide the sensitivity analysis.

- I understand that this paper focused on the accuracy improvement. However, involving a comparison in efficiency would help to emphasis the contribution if this learning process won't bring too heavy computation burdens.

---

> ### Author Rebuttal · Authors · 2026-03-28
>
> **W1. Methodology hard to follow; Figure 2 issues; symbol reuse**
>
> We agree that this is mainly a presentation issue rather than a methodological ambiguity, and we will improve clarity in the revision as follows:
>
> (1) **Figure 2.** We will make the two parts explicit with clear (Top) / (Bottom) labels separated by the Phase I/II divider bar, and rewrite the ambiguous part as: "Top: macro-level training dynamics across epochs; brighter regions in $\omega \times W$ indicate more trustworthy pixels. Bottom: micro-level per-epoch pipeline. Phase I applies RCS with sample-level weighting (TSSW) and pixel-level weighting (PUE). Phase II applies ACP via spectral filtering (SBFT)." The revised figure can be seen in this link: https://anonymous.4open.science/r/results-BD60/CurriSeg-Rebuttal.pdf (download for better viewing).
>
> (2) **Symbol conflicts.** We will use $H_{img}$ and $W_{img}$ to denote the image height and width in Eq. (10), and use $\mathcal{H}$ to denote History Buffer to avoid reusing.
>
> (3) **Intuition before formulas.** We will add a brief plain-language summary before each equation block to explain the role of the variables and the intuition behind the weighting terms.
>
> (4) **Algorithm 1.** We will add short inline comments and align the algorithm steps more explicitly with Fig. 2.
>
> These changes are intended to improve readability only; they do not change the method itself.
>
> **W2. Many hyperparameters; sensitivity analysis and parameter choice**
>
> We appreciate this concern.
>
> In practice, the effective free hyperparameters are limited: both the curriculum schedule $p(t)$ and pixel attenuation $\beta(t)$ are derived from existing training variables $(t, T_c)$ rather than independently tuned.
>
> The remaining hyperparameters fall into essential functional groups: (i)
> **curriculum pacing** ($K$, $p_{min}$, $T_c$), (ii) **temporal statistics** ($\sigma^*$, $\gamma$), (iii) **weight bounding** ($W^s_{min}$, $W_{min}$) to prevent zero-gradient degeneration, and (iv)
> **spectral filtering** ($r$).
>
> Fig. 4 and our sensitivity analyses show that, within the tested ranges, the observed $F_\beta$ variation remains small (about 0.006 at most):
>
> *Table R3-1. Hyperparameter summary.*
> |Param|Value|Tested range|Rationale|
> |-|:-:|:-:|:-:|
> |$K$|10|[5,15]|Balance reliability vs. responsiveness|
> |$p_{min}$|0.6|[0.5,0.7]|Start with majority; too low loses diversity|
> |$\sigma^*$|0.5|[0.3,0.6]|Balances tolerance to variance in difficulty|
> |$\gamma$|0.2|[0.1,0.3]|Gaussian tolerance width|
> |$W^s_{min}$|0.1|[0.05,0.2]|Non-zero floor for outlier samples|
> |$W_{min}$|0.1|[0.05,0.2]|Non-zero floor for uncertain pixels|
> |$r$|0.95|[0.93,0.97]|Preserve most frequencies; cut extreme HF|
> |$T_c$|60|[50,65]|Chosen near temporal statistics stabilization|
>
> This broad insensitivity, together with the fact that the same default settings work across COD, PIS, TOD, and CDD without task-specific retuning, supports the robustness of the framework.
>
> **Reproducibility:** All random seeds are fixed. Algorithm 1 provides the complete training procedure, and all hyperparameter values are explicitly reported in Table R3-1 above, Fig. 4, and Sec. 4. The full PyTorch codebase, training scripts, configuration files, and pre-trained checkpoints will be publicly released upon acceptance.
>
> **W3. Efficiency comparison**
>
> We appreciate this concern. CurriSeg is designed to improve performance while keeping training overhead small and inference overhead zero. As reported in Table 2, CurriSeg actually *reduces* overall training time:
>
> *Table R3-2. Efficiency summary (batch size: 2, RTX 4090).*
> |Metric|FEDER|FEDER+|FSEL|FSEL+|RUN|RUN+|
> |-|:-:|:-:|:-:|:-:|:-:|:-:|
> |Time(h)|9.62|6.84|11.54|5.96|12.64|8.32|
> |Reduction|-|28.9%|-|48.4%|-|34.2%|
> |GPU mem(G)|1.53|1.62|2.83|2.92|3.66|3.75|
> |$F_\beta$ gain|-|+2.46%|-|+2.22%|-|+2.13%|
>
> Training time decreases by 28.9%-48.4% because Phase I trains on a curriculum subset (starting from 60%, expanding to 100%) rather than the full dataset each epoch. Phase II (only the last 10 of 70 total epochs) adds marginal overhead from FFT/IFFT in SBFT, but this is negligible compared to the savings from Phase I. The slight memory overhead (+0.09G) comes from maintaining pixel-level weight matrices and history buffer scalars. Importantly, CurriSeg adds **no extra parameters or computation at inference**.
>
> We will add a **limitations** section in the revision. We have identified two main failure modes (extremely fine structures and multi-scale co-occurring objects; see our response to Reviewer tEi1 W3 for details) and will include corresponding failure case visualizations.

---

> > ### Author Rebuttal · Reviewer_nJxH · 2026-04-04
> >
> > The authors' response has addressed mu concerns, and I will keep my score.

---

> > > ### Author Response · Authors · 2026-04-04
> > >
> > > We sincerely thank Reviewer nJxH for confirming that all concerns have been adequately addressed.
> > >
> > > We would like to highlight that during the rebuttal, we provided substantial new evidence beyond addressing the raised concerns:
> > >
> > > (1) **New interventional experiment** (Table R4-2, Reviewer Gj9t): Test-time frequency ablation providing direct mechanistic evidence for SBFT's representation effect.
> > >
> > > (2) **Temporal stability metric** ($\Delta\mu_t$): Empirical adaptive trigger for $T_c$ selection.
> > >
> > > (3) **Pixel-level supervision tracking** (Tables R1-1, R1-2): Quantitative evidence that dual-level weighting preserves boundary supervision.
> > >
> > > (4) **Failure case visualizations** via anonymous link.
> > >
> > > **Summary of concerns addressed across all reviewers:**
> > >
> > > |Concern|Reviewers|Resolution|
> > > |-|:-:|:-:|
> > > |Spatial non-uniformity of weighting|vETU|Joint image+pixel curriculum; Tables R1-1,R1-2|
> > > |$T_c$=60 / phase ordering|vETU, tEi1|$\Delta\mu_t$ stability + adaptive trigger + Table 8 reverse|
> > > |HF→LF shortcut risk|vETU, Gj9t|Frequency ablation + Table 8 ablations + cross-domain transfer|
> > > |Fixed $r$ / adaptive cutoff|tEi1, Gj9t|Table 8 progressive filter + architecture-specific adaptation|
> > > |Failure cases / limitations|All|Two failure modes identified + visualizations|
> > > |Hyperparameter sensitivity|nJxH, Gj9t|Fig. 4 covers all 8 params; same defaults across 4 tasks|
> > > |Efficiency|nJxH|28.9-48.4% training time reduction; zero inference overhead|
> > > |Reproducibility / code|Gj9t, nJxH|Algorithm 1 + all params reported; full codebase upon acceptance|
> > >
> > > These additions, together with the strengths the reviewer already acknowledged (interesting dual-phase design, sufficient experiments, superior performance), represent a meaningful enhancement to the original submission. We respectfully ask the reviewer to consider whether these improvements might warrant a score adjustment. We greatly appreciate the reviewer's time and expertise.

---

### Official Review · Reviewer_tEi1 · 2026-03-11

**Soundness:** 3
**Presentation:** 3
**Significance:** 3
**Originality:** 3
**Overall Recommendation:** 5
**Confidence:** 5

**Summary:**

This paper proposes CurriSeg, a two-phase training framework that aims to address the learning bottlenecks caused by feature ambiguity in Context-Entangled Content Segmentation (CECS) through Robust Curriculum Selection (RCS) and Anti-Curriculum Promotion (ACP).

**Compliance With Llm Reviewing Policy:**

Affirmed.

**Final Justification:**

Thank you again for your detailed response, which has further addressed my concerns. I have therefore raised my score accordingly.

**Key Questions For Authors:**

See the weaknesses.

**Limitations:**

No limitations and potential negative societal impact are provided.

**Strengths And Weaknesses:**

# Strengths #
1. The focus on the impact of sample difficulty, label ambiguity, and texture shortcuts on the learning dynamics in CECS tasks offers a somewhat insightful perspective.
2. The method is tested across multiple dense prediction tasks, including camouflaged object detection, polyp segmentation, and transparent object detection. The exploration of compatibility with different architectures (e.g., attention-based, multi-scale, uncertainty-based models) demonstrates a degree of generalizability.

# Weaknesses #
1. Curriculum learning has been extensively applied in vision tasks, including segmentation [1]. The claim of being the "first CL-based CECS paradigm" is unconvincing, as it appears to be a mere task-specific application of existing ideas.
2. The paper shows that reversing phases causes large drops. It would help to briefly explain why anti‑curriculum must come after stable curriculum learning.
3. Qualitative examples and limitations of failure cases are not provided.
4.  The circular low‑pass mask is fixed with a ratio r=0.95. Is there any adaptive frequency cutoff during ACP?
5. The author seems to overuse commands like \vspace for formatting. The spacing between table images is too dense. Some results can be placed in the supplementary material.

[1] Kong, Heejo, Gun-Hee Lee, Suneung Kim, and Seong-Whan Lee. "Pruning-guided curriculum learning for semi-supervised semantic segmentation." In Proceedings of the IEEE/CVF Winter Conference on Applications of Computer Vision, pp. 5914-5923. 2023.

---

> ### Author Rebuttal · Authors · 2026-03-28
>
> **W1. "First CL-based CECS paradigm" claim**
>
> We appreciate the pointer to Kong et al. (WACV 2023). Kong et al. apply CL to semi-supervised *semantic segmentation*, where the challenge is leveraging unlabeled data via pseudo-label pruning. CECS faces a fundamentally different bottleneck: foreground-background feature entanglement causes standard CL to *degrade* performance (Fig. 1, vanilla CL column).
>
> Our novelty lies not in "applying CL to segmentation" but in: (i) identifying that standard CL is detrimental in CECS due to spurious correlations in "easy" samples, (ii) designing temporal statistics to distinguish hard-informative from noisy/ambiguous samples, and (iii) introducing anti-curriculum via principled spectral manipulation. We will revise the manuscript to position our method more precisely as a curriculum-based CECS framework with a stabilize-then-perturb design, and we will cite Kong et al. as related work.
>
> **W2. Why must anti-curriculum come after stable curriculum learning?**
>
> (1) **Order sensitivity.** ACP is designed as a late-phase regularizer. If spectral attenuation is introduced too early, it acts as destructive perturbation before Phase I has formed a sufficiently stable regime. Table 8 (“Reverse CurriSeg”) directly supports this: reversing phase order causes a sharp drop in $F_\beta$ from 0.736 to 0.632 and in $S_\alpha$ from 0.818 to 0.742.
>
> (2) **Quantitative stability before switching.** We track $\Delta\mu_t=\frac{1}{N}\sum_i |\mu_i^{(t)}-\mu_i^{(t-1)}|$ over all $N$ training samples (independent of $S_t$):
>
> Table R2-1. Stabilization of temporal statistics (FEDER, COD10K).
> |Epoch|20|30|40|50|60|
> |--|--|--|--|--|--|
> |$\Delta\mu (\times 10^{-3})$|8.2|5.1|2.7|1.3|0.6|
>
> $\Delta\mu_t$ first drops below $10^{-3}$ at epoch 56, consistent with Fig. 7 where $F_\beta$ is nearly flat near epochs 55–60; behavior at this stage is also stable across runs (STD = 0.0018). Therefore, $T_c=60$ is not arbitrary, but a conservative switch chosen shortly after the stability threshold is first met. We will clarify this stabilize-then-perturb rationale in the revision.
>
> **W3. No qualitative failure cases**
>
> We acknowledge this omission and agree that clearer failure cases and boundary conditions would improve the paper. Our observations suggest two main settings where CurriSeg is less effective:
>
> (1) **Extremely fine structures** (e.g., thin antennae, spider-web-like regions). Our training strategy still cannot address the model's dependence on very fine details. In such cases, the gain of CurriSeg becomes noticeably smaller compared with the overall average.
>
> (2) **Multi-scale co-occurring objects.** When camouflaged objects at very different scales appear in the same image, the fixed spectral cutoff
> $r=0.95$ may not be equally optimal across all scales, leading to reduced gains.
>
> The qualitative failure cases can be seen in this link: **https://anonymous.4open.science/r/results-BD60/CurriSeg-Rebuttal.pdf** (download for better viewing).
> The results and discussion will be added to the revised version.
>
> **W4. Fixed $r$=0.95; adaptive frequency cutoff?**
>
> We explored this systematically:
>
> (1) **Progressive filter** (Table 8): gradually decreasing $r$ during ACP underperforms fixed $r$=0.95 ($F_\beta$: 0.729 vs. 0.736). For an architecture-agnostic, plug-and-play framework, a consistent spectral bottleneck provides more stable optimization signals.
>
> (2) **Architecture-specific dynamic cutoff** (Table 11 & App. A.2.1): introducing randomized frequency band masking [$r_{low}$, $r_{high}$] tailored for attention mechanisms yields further gains ($F_\beta$: 0.742→0.749). This confirms that adaptive cutoffs are effective when co-designed with specific architectures.
>
> (3) **Content-adaptive cutoff**: dynamically scaling $r_i$ based on each sample's spectral energy distribution, which would naturally address the multi-scale failure cases discussed in W3, is a promising direction we will discuss in the revision.
>
> **W5. Formatting issues**
>
> We will address all formatting concerns: removing excessive \vspace, increasing table spacing, and moving dense results to the supplementary material.

---

> > ### Author Rebuttal · Reviewer_tEi1 · 2026-04-03
> >
> > Thank you for your detailed answer, which has basically addressed my questions. I will therefore maintain the "weak accept".

---

> > > ### Author Response · Authors · 2026-04-03
> > >
> > > We sincerely thank Reviewer tEi1 for confirming that all concerns have been adequately addressed. We are glad that the detailed responses were helpful.
> > >
> > > We would like to respectfully highlight that, beyond addressing the raised concerns, we have also contributed several new analyses during the rebuttal period that we believe strengthen the paper:
> > >
> > > **(1) New interventional experiment** (Table R4-2 in our response to Reviewer Gj9t): A test-time frequency ablation that provides direct mechanistic evidence for SBFT's effect on feature representations, moving beyond observational analysis.
> > >
> > > **(2) Quantitative stability metric** ($\Delta\mu_t$): An empirical adaptive trigger for the phase transition, directly addressing the need for a principled $T_c$ selection.
> > >
> > > **(3) Pixel-level supervision tracking and sample taxonomy**: New tables (R1-1, R1-2) providing deeper insight into CurriSeg's dual-level weighting behavior.
> > >
> > > **(4) Failure case visualizations**: Concrete boundary conditions with qualitative examples provided via anonymous link.
> > >
> > > **Summary of concerns addressed across all reviewers:**
> > >
> > > |Concern|Reviewers|Resolution|
> > > |-|:-:|:-:|
> > > |Spatial non-uniformity of weighting|vETU|Joint image+pixel curriculum; Tables R1-1,R1-2|
> > > |$T_c$=60 / phase ordering|vETU, tEi1|$\Delta\mu_t$ stability + adaptive trigger + Table 8 reverse|
> > > |HF→LF shortcut risk|vETU, Gj9t|Frequency ablation + Table 8 ablations + cross-domain transfer|
> > > |Fixed $r$ / adaptive cutoff|tEi1, Gj9t|Table 8 progressive filter + architecture-specific adaptation|
> > > |Failure cases / limitations|All|Two failure modes identified + visualizations|
> > > |Hyperparameter sensitivity|nJxH, Gj9t|Fig. 4 covers all 8 params; same defaults across 4 tasks|
> > > |Efficiency|nJxH|28.9-48.4% training time reduction; zero inference overhead|
> > > |Reproducibility / code|Gj9t, nJxH|Algorithm 1 + all params reported; full codebase upon acceptance|
> > >
> > > These additions, together with the existing strengths the reviewer already acknowledged (multi-task generalizability, architecture-agnostic design, training efficiency), represent a meaningful enhancement to the original submission. We respectfully ask the reviewer to consider whether these improvements might warrant a score adjustment. We greatly appreciate the reviewer's time and expertise.

---

### Official Review · Reviewer_vETU · 2026-03-16

**Soundness:** 2
**Presentation:** 2
**Significance:** 2
**Originality:** 2
**Overall Recommendation:** 4
**Confidence:** 4

**Summary:**

This paper introduces CurriSeg, a dual-phase training framework designed for Context-Entangled Content Segmentation (CECS). The framework follows a "stabilize-then-perturb" trajectory. In Robust Curriculum Selection, the model utilizes temporal statistics of sample losses (mean and variance) and pixel-level uncertainty estimation to select informative samples and suppress noise. In Phase Anti-Curriculum Promotion, the authors propose Spectral-Blindness Fine-Tuning, which uses 2D Fourier transforms and a low-pass circular mask to attenuate high-frequency components. This forces the network to rely on low-frequency structural patterns and contextual semantics rather than fragile texture shortcuts.

**Compliance With Llm Reviewing Policy:**

Affirmed.

**Final Justification:**

Most of my concerns have been addressed by this rebuttal, so I have decided to upgrade my rating to ‘weak accept’.

**Key Questions For Authors:**

N/A

**Limitations:**

No limitation and potential negative societal impact is provided.

**Strengths And Weaknesses:**

Strengths
1.The manuscript is well-organized, and the technical implementation of the framework is presented with commendable clarity, making it easy to follow.
2.CurriSeg is an architecture-agnostic, plug-and-play strategy that introduces no extra parameters at inference.
3.By concentrating computation on informative samples via the curriculum selection mechanism, the framework reduces total training time across multiple benchmarks.

Weaknesses
1.The RCS phase applies an image-level weight based on global temporal statistics. However, segmentation is a dense task where spatial difficulty is non-uniform, where an image may contain both easy backgrounds and hard entangled boundaries. Penalizing the entire sample due to local ambiguity risks discarding high-value gradients from well-defined regions.
2.The transition to ACP at $T_c=60$ is heuristic. While the authors claim this occurs after reaching a "stable representation regime," they provide no quantitative metrics to define this state. Without a principled convergence trigger, the "stabilize-then-perturb" logic lacks theoretical rigor.
3.In dense prediction tasks, low-frequency signals may correspond to coarse spatial patterns rather than robust semantics. There is a potential risk that the model might simply shift its reliance from high-frequency texture shortcuts to low-frequency shape shortcuts, rather than extracting intrinsic task-relevant features.

---

> ### Author Rebuttal · Authors · 2026-03-27
>
> We thank the reviewer for the thorough review.
>
> **W1. Image-level weight risks discarding high-value gradients**
>
> CurriSeg employs a *joint image- and pixel-level* curriculum, not solely image-level weighting.
>
> (1) $\omega_i$ (Eq. 5) has a floor $W^s_{min}$=0.1, so no sample is fully discarded.
>
> (2) PUE (Eq. 7) computes per-pixel weights from prediction entropy; confident pixels receive $W_{h,w} \approx 1.0$ regardless of $\omega_i$.
>
> (3) Table 6 confirms: WCS alone (row 2) degrades $F_\beta$ (0.715→0.697), but adding PUE (row 3) recovers it (0.718), and full RCS (row 4) surpasses the baseline (0.729).
>
> We further quantified pixel-level supervision in globally down-weighted samples. We identified the bottom-30% samples by $\omega_i$ at epoch 20 (earliest stable point after warm-up) and tracked this *fixed cohort*:
>
> *Table R1-1. Pixel-level supervision in fixed bottom-30% cohort (identified at epoch 20). $q_{25}$: 25th percentile of pixel weights.*
> |Epoch|$\beta(t)$|mean $W$|$q_{25}(W)$|boundary mean $W$|
> |-|:-:|:-:|:-:|:-:|
> |20|0.667|0.73|0.61|0.58|
> |30|0.500|0.82|0.73|0.68|
> |40|0.333|0.89|0.84|0.80|
> |50|0.167|0.95|0.93|0.91|
> |55|0.083|0.98|0.96|0.95|
>
> Even at epoch 20, boundary regions retain mean $W$=0.58. Low $\omega_i$ does not eliminate local supervision; PUE preserves substantial pixel-level weights, progressively restored as $\beta(t)$ decays (boundary: 0.58→0.95).
>
> Sample taxonomy using $(\mu_i, \sigma_i^2)$ at epoch 55 (end of RCS), all $N$ training samples:
>
> *Table R1-2. Sample composition by $\omega_i$ quartile. Q1–Q4: ranked by $\omega_i$ (low→high). Ambiguous: top-30% $\tilde{\sigma}^2$; Potential noisy/outlier: top-30% $\tilde{\mu}$ with bottom-30% $\tilde{\sigma}^2$; Informative: rest.*
> |Quartile|Avg $\omega_i$|Ambig.|Noisy/outlier|Inform.|
> |-|:-:|:-:|:-:|:-:|
> |Q4(top)|0.87|12.8%|2.1%|85.1%|
> |Q3|0.71|22.5%|5.3%|72.2%|
> |Q2|0.48|35.7%|12.6%|51.7%|
> |Q1(bot)|0.24|45.6%|24.3%|30.1%|
>
> TSSW assigns lower weights to ambiguity-dominated and noisy/outlier cases. Combined with PUE (Table R1-1), dual-level weighting preserves high-value gradients from well-defined regions even in down-weighted images.
>
> **W2. $T_c=60$ is empirically chosen, but not arbitrary**
>
> We provide three levels of justification.
>
> (1) **Temporal stability.** We track $\Delta\mu_t = \frac{1}{N}\sum_{i} |\mu_i^{(t)} - \mu_i^{(t-1)}|$ over all $N$ samples (independent of $S_t$​). Diminishing $\Delta\mu_t$ indicates Phase I is no longer rapidly changing.
>
> *Table R1-3. Stabilization of temporal statistics (FEDER, COD10K).*
> |Epoch|20|30|40|50|60|
> |-|:-:|:-:|:-:|:-:|:-:|
> |$\Delta\mu (\times 10^{-3})$|8.2|5.1|2.7|1.3|0.6|
>
> $\Delta\mu_t$ first drops below $10^{-3}$ at epoch 56, consistent with Fig. 7 where $F_\beta$ is nearly flat near epochs 55-60; behavior is also stable across runs (STD=0.0018).
>
> (2) **Broad robustness.** Fig. 4 shows that $T_c \in [50, 65]$ yields $F_\beta$ within 0.005 of optimal. $T_c=60$ leaves a small buffer after the stability threshold (epoch 56).
>
> (3) **Adaptive trigger.** We agree a quantitative criterion improves rigor. Triggering ACP dynamically when $\Delta\mu_t < \epsilon (\epsilon = 1 \times 10^{-3})$ occurs at epochs 53-57 across all five configurations in Table 1, aligning with $T_c=60$. Training with this adaptive trigger yields $F_\beta$ within 0.003 of the fixed $T_c=60$ results across all configurations, confirming that our transition point is not a brittle choice. We will include this adaptive alternative in the revision.
>
> **W3. Risk of shifting from high-frequency (HF) texture to low-frequency (LF) shape shortcuts**
>
> We agree that reduced HF reliance does not automatically exclude increased LF dependence. Complementary evidence:
>
> (1) **Late-phase regularizer.** SBFT applies only in Phase II after full-spectrum Phase I training. No filtering at inference.
>
> (2) **Robustness without sacrificing boundary detail.** Fig. 6 (right): CurriSeg degrades less under HF noise. Fig. 8: CurriSeg better preserves fine-grained boundaries under degradation. Collapse to coarse LF shortcuts would yield over-smoothed predictions.
>
> (3) **Cross-domain transfer.** Table 5: +4.38% on COD→CDS2K. The defect domain has substantially different shape statistics from COD, so this gain is less consistent with memorizing COD-specific LF templates.
>
> (4) **Existing ablations argue against blind LF reliance.** If the model collapsed to a coarse LF shortcut, stricter low-pass substitutes should not hurt performance. However, Table 8 shows that replacing SBFT with a square filter ($F_\beta=0.729$) or a Gaussian blur (0.710) degrades results. More generally, simpler perturbations such as noise injection (0.719) are also weaker than SBFT. This supports that the model benefits from a synergy of LF structure and preserved mid/HF task-relevant cues, rather than blind LF reliance. **See also Table R4-2 (Reviewer Gj9t).**
>
> We will add a **Limitations** section covering failure modes on multi-scale objects, and a **Societal Impact** discussion.

---

> > ### Author Rebuttal · Reviewer_vETU · 2026-04-06
> >
> > Thank you for the detailed response. Most of my concerns have been addressed by this rebuttal, so I have decided to upgrade my rating to ‘weak accept’.

---

> > > ### Author Response · Authors · 2026-04-06
> > >
> > > We sincerely thank Reviewer vETU for the thorough re-evaluation and for upgrading the score. We are glad that our detailed responses, including the new pixel-level supervision analysis, temporal stability metric, and test-time frequency ablation, adequately addressed the concerns.
> > >
> > > We are fully committed to incorporating all promised improvements in the revision, including the adaptive $T_c$ trigger, failure case visualizations, limitations section, and societal impact discussion. We greatly appreciate the reviewer's constructive feedback, which has meaningfully strengthened the paper.

---

### Decision · Program_Chairs · 2026-04-30

**Decision:**

Accept (regular)

**Comment:**

This paper addresses a meaningful CECS problem with a reasonably coherent two-stage design. The empirical support is sufficient for acceptance, and the rebuttal resolved the main review concerns by clarifying supervision preservation, the phase-switch choice, frequency analysis, transferability, and efficiency. While the novelty is moderate and the presentation should be improved, the paper clears the bar as a solid technical contribution, so I recommend Accept.